# Nordic Seaweed and Diabetes Prevention: Exploratory Studies in KK-Ay Mice

**DOI:** 10.3390/nu11061435

**Published:** 2019-06-25

**Authors:** Lasse E. Sørensen, Per B. Jeppesen, Christine B. Christiansen, Kjeld Hermansen, Søren Gregersen

**Affiliations:** 1Department of Endocrinology and Internal Medicine, Aarhus University Hospital, Palle Juul-Jensens Boulevard 99, 8200 Aarhus N, Denmark; elsoerensen@gmail.com (L.E.S.); kjeld.hermansen@aarhus.rm.dk (K.H.); 2Department of Clinical Medicine, Aarhus University, Palle Juul-Jensens Boulevard 165, 8200 Aarhus N, Denmark; per.bendix.jeppesen@clin.au.dk (P.B.J.); cbch93@gmail.com (C.B.C.); 3Steno Diabetes Center Aarhus, Palle Juul-Jensens Boulevard 165, 8200 Aarhus N, Denmark

**Keywords:** diabetes, KK-Ay mice, seaweed, algae, *Alaria esculenta*, *Saccharina latissima*, *Palmaria palmata*

## Abstract

Background: The global epidemic of type 2 diabetes (T2D) is a challenging health problem. Lifestyle changes, including nutrition therapy, areimportant for the prevention and management of T2D. Seaweeds contain several bioactive substances with potential health properties and may be a low-cost alternative functional food in the prevention of T2D. Objective: The aim of this study was to explore the preventive effects of dried Nordic seaweed species on diabetes in an animal model of T2D. Method: Fiftymale KK-Ay mice were randomly assigned to one of four diets: control diet (chow) or diets supplemented with *Alaria esculenta* (AE), *Saccharina latissima* (SL), or *Palmaria palmata* (PP). The effect of the interventions on the progression of T2D was monitored over 10 weeks and evaluated by circulating glucose, glycated hemoglobin (HbA1c), insulin, glucagon, and lipid levels. Results: The SL group had significantly lower bodyweight, lower HbA1c and insulin levels, as well as higher high density lipoprotein (HDL) cholesterol levels after the 10-week intervention than the control group. At the end of the study, the control group had significantly higher HbA1c (*p* < 0.001) than all of the seaweed groups. Conclusion: All seaweed groups improved HbA1C compared to control and *Saccharinalatissima* seaweed had concomitantly beneficial effects on glycemic control and lipid levels in KK-Ay diabetic mice.

## 1. Introduction

Many patients diagnosed with type 2 diabetes (T2D) meet the criteria for metabolic syndrome (MetS), which is a cluster of diseases consisting of hypertension, hyperglycemia, and dyslipidemia. Type 2 diabetes is characterized by elevated blood glucose due to absolute or relative insulin deficiency. The decrease in insulin secretion is caused by a reduction in beta-cell mass or impaired beta-cell function. Concomitantly, there is a hypersecretion of glucagon from pancreatic alphacells. The pathogenesis of insulin resistance and the development of T2D is complex. Genetic factors, diet, sedentary lifestyle, smoking, obesity, gluco- and lipotoxicity as well as inflammation play a role. T2D/MetS are important risk factors for cardiovascular disease and mortality. Fortunately, T2D/MetS progression can be counteracted by lifestyle changes (e.g., a healthy diet) [1,2,3,4,5].

There is a growing interest in exploring the effects of natural food products for the prevention of lifestyle-related diseases. Seaweed has previously been part of the Northern European diet, especially in Ireland, Scotland, Norway, and Iceland. Today, seaweed constitutes a substantial part of the diet in many East Asian countries. Studies suggest that seaweed consumption may positively influence risk markers of T2D and T2D progression [6,7]. Seaweeds contain several bioactive substances like polysaccharides, proteins, lipids, polyphenols, and pigments, all of which may have beneficial health properties [8]. Alginate, a polysaccharide distributed in the cell wall of brown algae, has beneficial effects on glucose metabolism [9]. Fucoidan, a sulfated polysaccharide also found in various species of brown algae, lowers alpha-glucosidase activity in vitro [10], blood glucose in db/db mice [11], and glycated hemoglobin and glucagon-like peptide-1 in type 2 diabetes patients [12]. The phlorotannin, dieckol, isolated from brown seaweed, was found to inhibit alpha-glucosidase activity in vitro [13], and to protect against the glucotoxicity-induced oxidative stress associated with diabetes [14,15], delay T2D development in a db/db mouse model [16], and reduce postprandial hyperglycemia in prediabetic subjects [17]. Phlorotannin possesses an inhibitory effect on human salivary alpha-amylase, which may be useful as a natural nutraceutical to prevent diabetes [18]. In addition, an antidiabetic effect of polyphenolic-enriched fractions mediated via alpha-glucosidase activity from brown algae has previously been demonstrated [19,20]. In addition, quercetin, a flavonoid found in fruits, vegetables, and seaweeds [21], improves glycemia and dyslipidemia in type 2 diabetic db/db mice [22] and inhibits tumor necrosis factor alpha-induced insulin-resistance in skeletal muscle cells [23]. The color of brown algae results from the dominance of the carotenoid fucoxanthin. Fucoxanthin has been found to improve insulin resistance, decrease blood glucose levels, and reduce cytokine production in adipose tissue. Interestingly, fucoxanthin promotes translocation and induction of glucose transporter 4 in the skeletal muscles of diabetic/obese KK-Ay mice. Studies also reveal that fucoxanthin induces mitochondrial uncoupling protein 1 (UCP1) in abdominal white adipose tissue mitochondria, leading to oxidation of fatty acids and heat production [24,25,26,27]. Furthermore, fucoxanthin lowers glycated hemoglobin in healthy subjects. This decline was more pronounced in subjects with a certain UCP1 genotype that has been shown to be a predisposing factor for obesity [28].

Seaweed from the Nordic regions may be a potential low-cost alternative functional food for the prevention of T2D. The macro- and micronutrient composition of seaweed positions it as a promising natural food supplement and source for extraction of bioactive compounds. The action of Nordic seaweed on diabetes prevention has, to our knowledge, not been studied before. We hypothesized that dried Nordic seaweed protects against diabetes. We aimed to study and compare the impact of a 10-week supplementation with the following Nordic seaweed species: *Alaria esculenta*, *Saccharina latissima*, and *Palmaria palmata* with chow food (control) on diabetes development in KK-Ay mice.

## 2. Materials and Methods

Male KK-Ay/Ta Jcl (genetic obese T2D) mice from Taconic Europe A/S (Ejby, Denmark) were used for this study. The mice were 5 weeks old at the time of delivery.

### 2.1. Seaweed Diets

The seaweed was delivered by Icelandic Blue Mussel & Seaweed (Stykkishólmur, Iceland). Twenty percent (weight-percent) of the dried seaweed was incorporated into pellets (Altromin type 1324, Lage, Germany). The control diet was made from the same batch. The pellets with incorporated seaweed were color coded to ensure separation of experimental diets. Three different types of seaweed diets were used: *Alaria esculenta*, *Saccharina latissima*, or *Palmaria palmata* and control diet. The macronutrient and iodine content of the different diets were provided by the manufacturers (Table 1). The macronutrient and trace element composition of the diets were derived from the information given by the producer (for *Alaria esculenta* and *Saccharina latissima*, Icelandic Blue Mussel & Seaweed (Stykkisholmur, Iceland)) and from Mouritsen et al. [29] and the mice were fed a standard chow diet (Altromin 1324, Brogaarden, Lage, Germany) until start of the intervention.

The final diets used in the intervention were analyzed by the Danish Veterinary and Food Administration (Lystrup, Denmark) for fat, protein, carbohydrate, fiber, and energy content (Table 2). All chemical analyses were performed on freeze-dried materials. Dry matter was determined by drying to constant weight at 103 °C for 20 h. Nitrogen was analyzed by the principle of DUMAS [30], and protein content was calculated as N × 6.25. Fat was extracted with diethyl ether after hydrogen chloride (HCl, Noida, India) hydrolysis according to the Stoldt procedure [31]. Total energy was calculated as the content of protein, carbohydrate, and fat by using the following conversion factors: 17, 17, and 37, respectively. The different seaweed types were mixed with 80% rat chow (Altromin 1324, Altromin GmbH, Lage, Germany), adjusted so that the vitamin and mineral contents were the same as in 100% Altromin 1324, and pelleted afterwards (Brogaarden, Lynge, Denmark).

### 2.2. Design

The study included 50 male KK-Ay mice. After one week of acclimatization, the animals (now age 6 weeks) were randomly assigned (*n* = 12–13 in each group) to one of the three experimental diets supplemented with either *Alaria esculenta* (*Alaria*), *Saccharina latissimi* (*Saccharina*), or *Palmaria palmata* (*Palmaria*) for 10 weeks. Regular chow served as control.

The animals were caged with two or three mice in each cage with free access to tap water and the abovementioned diets. The animal facility had temperature controlled at 22–24 °C and a 12 h light/dark cycle. The experiment was approved by the Danish Animal Ethics Council (approved 8 May 2015, No. 2015-15-0201-00592). At intervention week 0, tail blood was sampled. The blood was collected in chilled tubes containing a 3 µL mix of heparin and aprotinin (7.7 mg/mL aprotinin, 2.300 IU/mL heparin). The samples were immediately centrifuged (4.000 rpm, 10 min at 4 °C) and plasma was collected and frozen for subsequent analyses of glucose, insulin, glucagon, total-cholesterol, high density lipoprotein (HDL), low density lipoprotein (LDL), and triglycerides. Whole blood was collected separately in ethylenediaminetetraacetic acid-preserved collection tubes for glycated hemoglobin analysis.

During the 10 weeks of the intervention study, weight and whole blood glucose levels were measured every second week. Whole blood glucose was determined after an overnight fast of approximately 12 h and analyzed on a glucose meter (OneTouch, Accu-Chek Aviva, Roche Diagnostics A/S, Hvidovre, Denmark).

After 10 weeks, the animals were anesthetized (after overnight fast) using pentobarbital (2 g/kg bodyweight) injected intraperitoneally, and a large blood sample was collected from the retrobulbar plexus.

### 2.3. Oral Glucose Tolerance Test

At intervention week 9, an oral glucose tolerance test (OGTT) was performed after a 12-h fasting period. Glucose was administered by gavage (2 g D-glucose/kg bodyweight, as a 60% glucose solution). Whole blood glucose was measured at the following time-points: −15, 0 (immediately before the glucose infusion), 15, 30, 60, 90, 120, 180, and 240 min.

### 2.4. Analytical Procedures and Statistics

All analytical procedures were carried out in accordance with the manufacturer’s instructions. The plasma insulin concentrations at the start and end of the intervention were determined with a sensitive rat insulin RIA kit (Linco Research Inc., St Charles, MO, USA). Plasma glucagon was also analyzed by a RIA kit (Linco Research Inc., St Charles, MO, USA). The glucagon antibody was specific for pancreatic glucagon and has no cross-reaction with other glucagon-like peptides. Plasma glucose was measured using an enzymatic reference method (Roche, Basel, Switzerland) on the CobasC111 system (Roche Diagnostics, Basal, Switzerland) and whole blood was measured during the OGTT using a glucose meter (Accu.Chek Aviva, Roche Diagnostics Denmark, Hvidovre, Denmark). Glycated hemoglobin was measured using the Tina-quant II immune turbidimetric method on the CobasC111 system. This determination is based on the turbidimetric inhibition immunoassay for hemolyzed whole blood. In the Tina-quant method, the antibody recognizes the first three amino acids of the hemoglobin beta chain N-terminal [32], Val-His-Met, which are identical in *Homo sapiens* (Accession No.: P68871) and *Mus musculus*(Accession No. beta chain 1: P02088, Accession No. beta chain 2: P02089). The total-, HDL-, and LDL-cholesterol as well as triglyceride contents of the plasma samples were analyzed using an enzymatic colorimetric method (Roche, Basel, Switzerland), also on the CobasC111 system. The plasma thyroid stimulating hormone (TSH) concentrations at the start and end of the intervention were determined with a mouse TSH kit (TSZ ELISA, BiotangInc., Waltham MA, USA). 

All statistical calculations were performed using GraphPad Prism 4 (GraphPad Software Inc., La Jolla, CA, USA). Bartlett’s test was used to test for homogeneity of variances within groups. One-way-ANOVA was used to examine the overall differences between groups at each timepoint (start and end). When the *p*-value from the ANOVA test showed significant differences, a post hoc multiple comparison test, Bonferroni’s test, was used to identify the significant differences between groups. All figures were produced using GraphPad Prism 4.

## 3. Results

### 3.1. Animal Bodyweight

All groups had similar mean bodyweight at the beginning of the experiment. As shown in Table 3, the *Saccharina* group gained less weight compared to the other groups. Thus, from week 3 and onwards, there was a significant difference between the *Saccharina* group and the control group. At week 11, the *Saccharina* group had a significantly lower bodyweight compared to all of the other groups. The bodyweights of the other two groups did not differ significantly from the control. Due to urgent rescheduling of the OGTT, only half of the animals were weighed in the fasting state at week 9, and consequently data from week 9 were excluded.

### 3.2. Plasma Glucose

Fasting plasma glucose concentrations were measured from the samples taken at the beginning and end of the intervention. No significant differences were found between groups either at the start (data not shown) or at the end of the intervention (Figure 1).

### 3.3. Oral Glucose Tolerance Test

The area under the curves (AUCs) and incremental AUCs (iAUCs) of the glucose response during OGTTs were not significantly different between groups, as illustrated in Table 4.

### 3.4. Glycated Hemoglobin

As expected, there were no significant differences in levels of glycated hemoglobin between groups before the intervention (data not shown). At the end of the dietary intervention, the control group had significantly higher levels of glycated hemoglobin (*p <* 0.001) compared to all of the seaweed groups (Figure 2).

### 3.5. Plasma Insulin and Glucagon

Before the intervention, plasmainsulin levels were similar in the four groups (data not shown). After the intervention, the *Saccharina* group had significantly lower insulin levels (*p <* 0.01) (Figure 3a). No significant differences in plasma glucagon were found between groups either at the start (data not shown) or at the end of the intervention (Figure 3b).

### 3.6. Circulating Cholesterol and Triglyceride

No significant differences in circulating total cholesterol, LDL cholesterol, and triglycerides were found between groups ateither the start (data not shown) or after the intervention (Figure 4a,b,d, respectively). The level of HDL cholesterol was significantly higher in the *Saccharina* group than in the control group (Figure 4c).

### 3.7. Thyroid Stimulating Hormone

There were no significant differences in TSH levels between groups (before and after) (data not shown).

## 4. Discussion

The objective of our study was to explore the preventive effects of diets supplemented with three dried seaweed species on the development of diabetes in the diabetic mice model KK-Ay. We tested the impact of two brown algae (*Alariae sculenta* and *Saccharina latissimi*) and a red algae (*Palmaria palmata*). Our main finding was a lower glycated hemoglobin in all seaweed groups, and in the *Saccharina* group lower insulin levels and higher HDL levels were observed compared to the control group. Thus, all of the dried seaweed types studied showed positive long-term effects on glycemic control in our KK-Ay mice. This result corroborated previous studies showing beneficial effects of seaweed in db/db mice and type 2 diabetic patients [7,11,20].

The lower fasting insulin levels in the *Saccharina* group points towards improved insulin sensitivity. However, despite clear evidence for improved long-term glycemia in all seaweed groups, we were not able to show differences in the fasting glucose, AUCs, or iAUCs during an OGTT. The reason for this discrepancy is presently not clear. We cannot exclude that stress in relation to the OGTT could have diluted differences in blood glucose in the mice. However, several studies have reported an inhibitory effect of seaweed and bioactive substances from seaweed on alpha-glucosidase [19,20]. The lower fasting p-insulin in the *Saccharina* group compared to the control may indicate improved insulin sensitivity in the *Saccharina* group. This corroborates other studies on carotenoids from brown algae [24,25,26,27]. The lower bodyweight in the SL group may at least in part explain the improved insulin sensitivity in the SL group. We detected no significant changes in fasting p-glucagon levels between the seaweed and control groups.

When interpreting the metabolic data, one needs to consider the impact of the dietary supplements on food intake and bodyweight. The different types of seaweed supplements incorporated into the pellets affected the texture of the pellets differently. Thus, the pellets had a tendency to crumble, which severely influenced our registration of food intake. For this reason, we were unable to measure foodintake accurately, and the data has been excluded. At week 11, the *Saccharina* group had a significantly lower bodyweight compared to all of the other groups. The lower bodyweight in the *Saccharina* group may be explained by a significantly lower food intake in this group, which may indicate a higher degree of satiety. Another possibility could be increased weight loss due to oxidation of fatty acids, since studies have revealed that fucoxanthin, a pigment in brown algae, induces UCP1 in abdominal white adipose tissue mitochondria, leading to oxidation of fatty acids and heat production [24,25,26,27]. This interesting possibility needs to be explored. Seaweed can have higher mineral content, including iodine, than traditional vegetables. However, in the present study we found no significant differences in TSH between the intervention groups and the control, and consequently it seems unlikely that disturbances in thyroid function can explain the lower bodyweight in the SL group.

The essential amino acid score is generally higher in seaweed proteins than that of proteins in cereals and vegetables. However, the digestibility of seaweed proteins is not well documented and studies on their bioavailability in humans are scarce [8]. It has been suggested that small peptides from seaweed may possess bioactivity, for example, of relevance for blood pressure regulation [33]. In this study, we were unable to measure blood pressure, but another component of relevance for metabolic health is the observed increase in HDL cholesterol in the *Saccharina* group. Administration of fucoidan purified from seaweed to high fat-fed mice improved both total and HDL-cholesterol as well as triglycerides in a dose-dependent manner [34]. A former study also showedthat seaweed supplementation significantly increased HDL cholesterol [7], which is associated with a lower cardiovascular risk. Furthermore, seaweed may have protective effects against diabetes complications, since fucoidan extracted from seaweed prevents diabetic cardiomyopathy and reduces both interstitial fibrosis and myocyte hypertrophy in type 2 diabetic Goto–Kakizaki rats [35].

The major goal in the prevention and managementof T2D is to protect patients from associated long-term complications. Because insulin resistance plays a fundamental role in the pathogenesis of T2D, interventions aimed towards improvement in insulin sensitivity should play a critical role. Medications may have unwanted side-effects and although lifestyle changes can be difficult to maintain over long-term periods, exercise and diet still remain the key factors for the prevention and treatment of T2D. Our study indicates that some seaweed species have preventive effects on the development of T2D in KK-Ay mice. Moving forward, we need more knowledge about the chemical composition of Nordic seaweed and its digestion in animals as well as humans to obtain a more extensive understanding of the molecular mechanisms of action in the seaweed.

## Figures and Tables

**Figure 1 nutrients-11-01435-f001:**
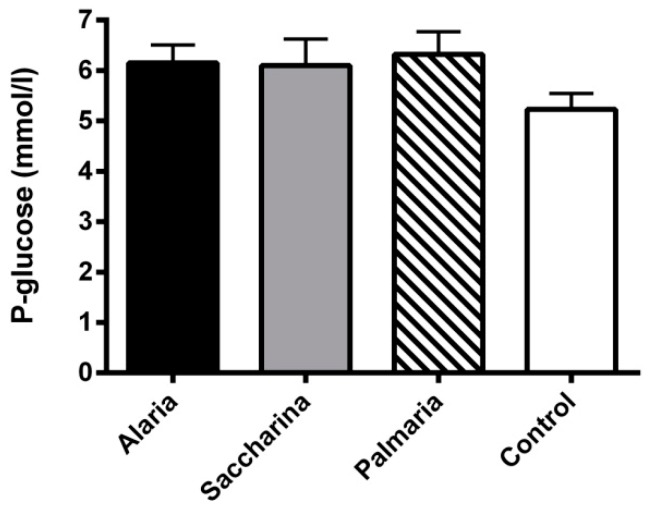
Fasting plasma glucose from the KK-Ay mice at the end of the 11-week intervention with diets supplemented with *Alaria esculenta*, *Saccharina latissima*, or *Palmaria palmata* compared to chow (control). No significant differences were found between groups. Values are means ± SEM.

**Figure 2 nutrients-11-01435-f002:**
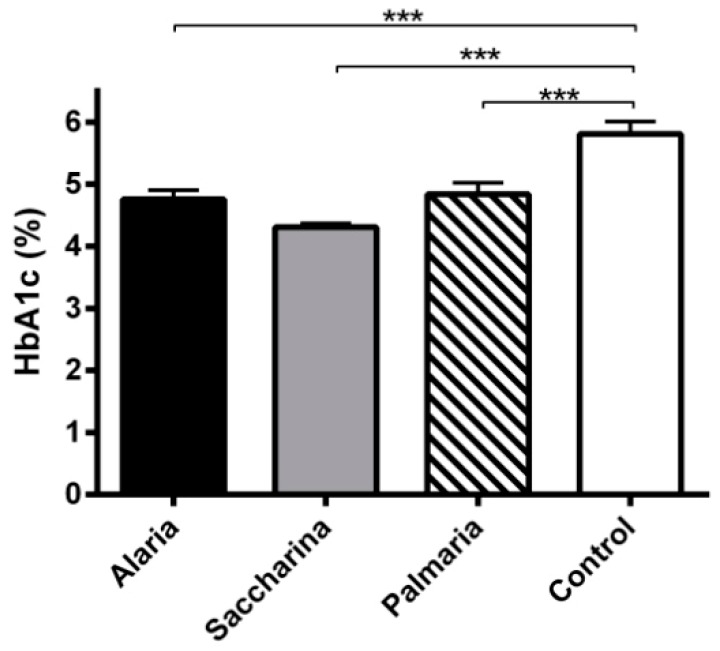
Glycated hemoglobin (HbA1c) from the KK-Ay mice at the end of the 11-week intervention with diets supplemented with *Alariaesculenta*, *Saccharinalatissima*, or *Palmariapalmata* compared to chow (control). The control group had significantly higher glycated hemoglobin levels compared to all of the seaweed groups *** (*p* < 0.001). Values are means ± SEM.

**Figure 3 nutrients-11-01435-f003:**
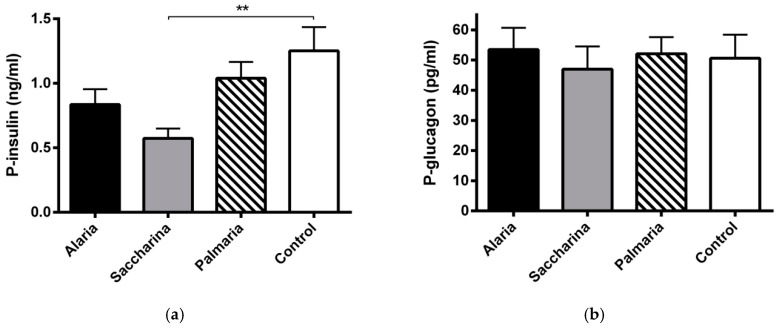
(**a**) Plasma insulin and (**b**) glucagon levels from the KK-Ay mice at the end of the 11-week intervention with diets supplemented with *Alaria esculenta*, *Saccharina latissima*, or *Palmaria palmata* compared to chow (control). The *Saccharina* group had significantly lower insulin levels compared to control ** (*p* < 0.01). No significant differences were found for plasma glucagon levels between groups. Values are means ± SEM.

**Figure 4 nutrients-11-01435-f004:**
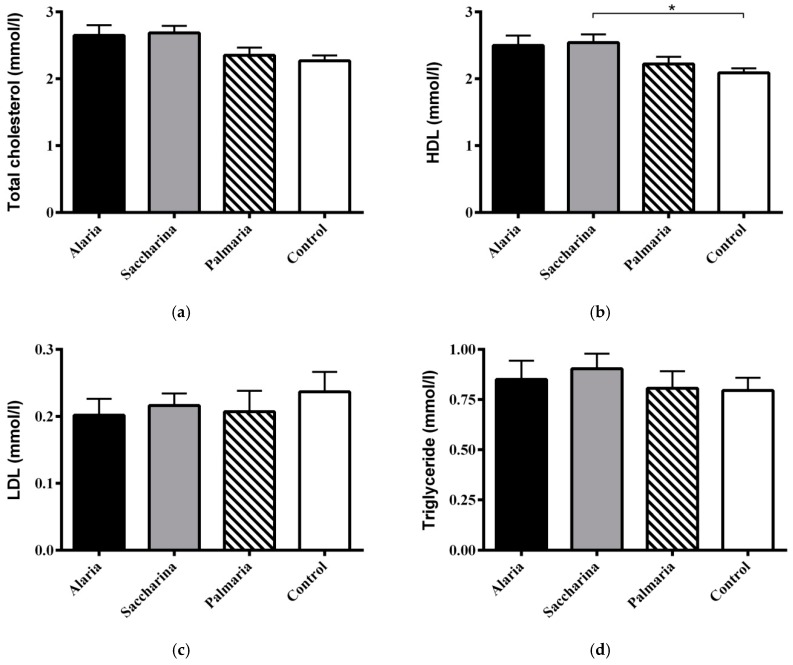
(**a**) Total plasma cholesterol, (**b**) HDL cholesterol, (**c**) LDL cholesterol, and (**d**) plasma triglyceride levels from the KK-Ay mice at the end of the 11-week intervention with diets supplemented with *Alaria esculenta*, *Saccharina latissimi*, or *Palmaria palmata* compared to chow (control). No significant differences were found for total and LDL cholesterol and triglyceride levels. Plasma HDL was higher for the *Saccharina* group compared to control * (*p* < 0.05). Values are means ± SEM.

**Table 1 nutrients-11-01435-t001:** The nutritional composition of the three different pure seaweed varieties.

	*Alaria esculenta*	*Saccharina latissima*	*Palmaria palmata*
Carbohydrates %	49.6	48.9	45
Protein %	16.1	11.4	20
Fat %	0.5	0.2	1
Iodine mg/100g	43	110	5–15

**Table 2 nutrients-11-01435-t002:** The nutritional composition of the four intervention diets. Twenty percent (weight-percent) dried seaweed was mixed with 80% chow (Altromin type 1324, Lage, Germany). The control diet was 100% chow made from the same batch.

	*Alaria esculenta*	*Saccharina latissima*	*Palmaria palmata*	Control
Energy KJ/100 g	1324	1301.6	1326	1385.9
Carbohydrates %	52.3	52.2	51.4	53.4
Protein %	18.4	17.4	19.2	19.2
Fat %	3.3	3.2	3.4	4.1

**Table 3 nutrients-11-01435-t003:** Bodyweight in grams (g) at week 1, 3, 5, 7, and 11 of the KK-Ay mice measured fed diets supplemented with *Alaria esculenta*, *Saccharina latissima*, *Palmaria palmata*, or chow. Values are means ± SD. * (*p* < 0.05), ** (*p* < 0.01), *** (*p* < 0.001) compared to control.

	*Alaria*	*Saccharina*	*Palmaria*	Control
Week 1	33.8 ± 2 g	33.6 ± 2 g	33.4 ± 2 g	33.7 ± 2 g
Week 3	36.5 ± 3 g	33.6 ± 3 g *	36.2 ± 5 g	37.5 ± 2 g
Week 5	38.9 ± 4 g	33.6 ± 3 g **	37.2 ± 5 g	39.8 ± 3 g
Week 7	39.3 ± 3 g	34.3 ± 3 g ***	37.9 ± 5 g	40.5 ± 3 g
Week 11	39.3 ± 3 g	35.8 ± 3 g ***	40.5 ± 3 g	41.5 ± 3 g

**Table 4 nutrients-11-01435-t004:** Area under the curves (AUCs) and incremental AUCs (iAUCs) during an oral glucose tolerance test (OGTT) in the KK-Ay performed at week 11 of the intervention with diets supplemented with either *Alariaesculenta*, *Saccharinalatissima,* or *Palmaria palmata* compared to chow (control). No significant differences between groups were found. Means ± SEM are shown.

	*Alaria*	*Saccharina*	*Palmaria*	Control
iAUC (mmol/L × 240 min)	1402 ± 205	1112 ± 164	1386 ± 190	1156 ± 121
AUC (mmol/L × 240 min)	2805 ± 270	2791 ± 254	3009 ± 251	2480 ± 123

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
