# Peer review of "Nordic Seaweed and Diabetes Prevention: Exploratory Studies in KK-Ay Mice"

_nutrients, 2019, doi:10.3390/nu11061435_

Round 1
Reviewer 1 Report
In this manuscript titled, "Nordic Seaweed and diabetes prevention: Exploratory studies in KK-Ay mice", Lasse E Sørensenet al., authors investigate the preventive effects of diets supplemented with three dried seaweed species on the development of diabetes in the diabetic mice model KK-Ay. Authors revealed that lower glycated hemoglobin in all seaweed groups and in the Saccharina group lower insulin levels and higher HDL levels than in the control group. All of the dried seaweed types studied showed positive long-term effects on glycemic control in KK-Ay mice. For the study, the manuscript is written clearly, however, the manuscript appears preliminary.
1. In this study, the authors didn’t reveal the mechanism of the preventive effects of diets supplemented with seaweed species. Are there any changes of UCP1in these mice at the mRNA and protein level?
2. Type 2 diabetic patients are at increased risk of cardiomyopathy and heart failure. Cardiomyopathy in diabetes is associated with a cluster of features including decreased diastolic compliance, interstitial fibrosis, and myocyte hypertrophy. Are there any changes of interstitial fibrosis and myocyte hypertrophy in these mice?
3. The p-value: p should be upper case and italic ‘ P ’, in this manuscript authors use both P and p.
4. In Figure 3 and 4, authors should improve the quality of all the images.
Author Response
To
Reviewer 1
Thank you for your comments. The manuscript has been revised according to your comments including revision of the introduction and discussion.
Comment 1: In this study, the authors didn’t reveal the mechanism of the preventive effects of diets supplemented with seaweed species. Are there any changes of UCP1in these mice at the mRNA and protein level?
Answer: KK-Ay mice express UCP1 in white adipose tissue on both protein and mRNA level1. As already stated in the manuscript, "studies reveal that fucoxanthin induces mitochondrial uncoupling protein 1 (UCP1) in abdominal white adipose tissue mitochondria, leading to oxidation of fatty acids and heat production [20-23]." Thus, increase in UCP1 could explain that the seaweed groups gain less weight than the control group. However, such analysis were not performed in the present study.
Comment 2: Type 2 diabetic patients are at increased risk of cardiomyopathy and heart failure. Cardiomyopathy in diabetes is associated with a cluster of features including decreased diastolic compliance, interstitial fibrosis, and myocyte hypertrophy. Are there any changes of interstitial fibrosis and myocyte hypertrophy in these mice?
Answer: Thank you for the comment: The effect of seaweed on cardiomyopathy has not been investigated in this study. However, seaweed might have protective effects and evidence suggests that fucoidan extracted from seaweed prevents diabetic cardiomyopathy and reduces both interstitial fibrosis and myocyte hypertrophy in type 2 diabetic Goto-Kakizaki rats2.We do not know if KK-Ay mice develop cardiomyopathy, and it is likely that another model should be applied to study the impact of seaweed on cardiomyopathy. This has not been a focus in this study.
Comment 3: The p-value: p should be upper case and italic ‘ P ’, in this manuscript authors use both P and p.
Answer: The manuscript has all p-values changed accordingly.
Comment 4: In Figure 3 and 4, authors should improve the quality of all the images.
Answer: The image resolution of the figures have been corrected.
On behalf of the authors,
Sincerely
Christine Bodelund Christiansen
Research assistant, Aarhus University
Palle Juul-Jensens boulevard 165
8200 Aarhus
Denmark
E-mail:CHBOCR@rm.dk, Mobile:+45 28518032
1Hayato M, Masashi H, Tokutake S, Katsura F, Kazuo M. Fucoxanthin from edible seaweed, Undaria pinnatifida, shows antiobesity effect through UCP1 expression in white adipose tissues. BiochemBiophys Res Commun 2005, 332, 392. https://doi.org/10.1016/j.bbrc.2005.05.002.
2Xinfeng Y, Quanbin Z, Wentong C. Low Molecular Weight Fucoidan Alleviates Cardiac Dysfunction in Diabetic Goto-Kakizaki Rats by Reducing Oxidative Stress and Cardiomyocyte Apoptosis. Journal of Diabetes Research 2014, 2014. https://doi.org/10.1155/2014/420929.

Reviewer 2 Report
The work of Soerensen et al is certainly of interest to the readers of Nutrition. Their work is original and the findings are well documented.
I have only minor questions:
1. Page 4, line 134. The measurement of glycated hemoglobin was performed by using an immunochemical method with antibodies prepared against human hemoglobin. Which is the evidence supporting the use of this method for mice’s hemoglobin? The authors should comment on this point, also at the light that mice hemoglobin is very different form man’s hemoglobin.
2. The same criticism applies to the determination of insulin and glucagon.
3. Table 3. Please report the units (grams?).
Author Response
To
Reviewer 2
Thank you for your useful comments. The manuscript have been updated according to the comments and an update of the introduction and discussion have been made. We here provide answers to the comments and suggestions:
Comment 1: Page 4, line 134. The measurement of glycated hemoglobin was performed by using an immunochemical method with antibodies prepared against human hemoglobin. Which is the evidence supporting the use of this method for mice’s hemoglobin? The authors should comment on this point, also at the light that mice hemoglobin is very different form man’s hemoglobin.
Answer: In the Tina-quant method, the antibody recognizes glucose as well as the first three amino acids of the hemoglobin beta chain N-terminal1, Val-His-Met, which are identical in both Homo sapiens (Accession No.: P68871) and Mus musculus (Accession No. beta chain 1: P02088, Accession No. beta chain 2: P02089).
Comment 2: The same criticism applies to the determination of insulin and glucagon.
Answer: These measurements have been performed using specific and commercially available kits that are widely used and reported in manuscripts2-6. The kits are validated for use on mice by the supplier.
Comment 3: Table 3. Please report the units (grams?).
Answer: The units are grams. This has been added to Table 3. Thank you.
On behalf of the authors,
Sincerely
Christine Bodelund Christiansen
Research assistant, Aarhus University
Palle Juul-Jensens boulevard 165
8200 Aarhus
Denmark
E-mail: CHBOCR@rm.dk, Mobile: +45 28518032
1John G. Glycated hemoglobin analysis. Ann Clin Biochem 1997, 34, 17. https://doi.org/10.1177/000456329703400105.
2 Miles D, Barak Y, He W, Evans RM, Olefsky M. Improved insulin-sensitivity in mice heterozygous for PPAR-gamma deficiency. J Clin Invest 2000,105, 287. https://doi.org/10.1172/JCI8538.
3Dludla PV, Gabuza KB, Muller CJF, Joubert E, Louw J, Johnson R. Aspalathin, a C-glucosyl dihydrochalcone from rooibos improves the hypoglycemic potential of metformin in type 2 diabetic (db/db) mice. Physiol Res 2018,14, 813.
4Tweedell A, Mulligan KX, Martel JE, Chueh FY, Santomango T, McGuinness OP. Metabolic response to endotoxin in vivo in the conscious mouse: role of interleukin-6. Metabolism 2011, 60, 92. https://doi.org/10.1016/j.metabol.2009.12.022
5 Boushey R, Abadir A, Drucker D, Hypoglycemia, defective islet glucagon secretion, but normal islet mass in mice with a disruption of the gastrin gene. Gastroenterology 2003, 125, 1164. https://doi.org/10.1016/S0016-5085(03)01195-8.
6Vuguin PM, Kedees MH, Cui L, et al. Ablation of the glucagon receptor gene increases fetal lethality and produces alterations in islet development and maturation. Endocrinology 2006,147, 3995. https://doi.org/10.1210/en.2005-1410

Round 2
Reviewer 1 Report
In this manuscript titled, "Nordic Seaweed and diabetes prevention: Exploratory studies in KK-Ay mice", Lasse E Sørensenet al., authors investigate the preventive effects of diets supplemented with three dried seaweed species on the development of diabetes in the diabetic mice model KK-Ay. Authors revealed that lower glycated hemoglobin in all seaweed groups and in the Saccharina group lower insulin levels and higher HDL levels than in the control group. All of the dried seaweed types studied showed positive long-term effects on glycemic control in KK-Ay mice. The manuscript is written clearly and enough improved, for the study the presented data are quite sufficient.